# Miro2 tethers the ER to mitochondria to promote mitochondrial fusion in tobacco leaf epidermal cells

Rhiannon R. White [1], Congping Lin[2,4,5], Ian Leaves [1], Inês G. Castro [1,6], Jeremy Metz [1], Benji C. Bateman [3], Stanley W. Botchway [3], Andrew D. Ward [3], Peter Ashwin [2] & Imogen Sparkes [1,7✉]

Mitochondria are highly pleomorphic, undergoing rounds of fission and fusion. Mitochondria are essential for energy conversion, with fusion favouring higher energy demand. Unlike fission, the molecular components involved in mitochondrial fusion in plants are unknown. Here, we show a role for the GTPase Miro2 in mitochondria interaction with the ER and its impacts on mitochondria fusion and motility. Mutations in AtMiro2's GTPase domain indicate that the active variant results in larger, fewer mitochondria which are attached more readily to the ER when compared with the inactive variant. These results are contrary to those in metazoans where Miro predominantly controls mitochondrial motility, with additional GTPases affecting fusion. Synthetically controlling mitochondrial fusion rates could fundamentally change plant physiology by altering the energy status of the cell. Furthermore, altering tethering to the ER could have profound effects on subcellular communication through altering the exchange required for pathogen defence.

[1] Biosciences, CLES, Exeter University, Exeter EX4 4QD, UK. [2] Department of Mathematics, Harrison Building, University of Exeter, Exeter EX4 4QF, UK. [3] Central Laser Facility, Science and Technology Facilities Council, Research Complex at Harwell, Didcot, Oxon OX11 0FA, UK. [4] Present address: Center for Mathematical Sciences, Huazhong University of Science and Technology, Wuhan, China. [5] Present address: Hubei Key Lab of Engineering Modeling and Scientific Computing, Huazhong University of Science and Technology, Wuhan, China. [6] Present address: Department of Molecular Genetics, Weizmann Institute of Science, Rehovot 7610001, Israel. [7] Present address: School of Biological Sciences, University of Bristol, Bristol Life Sciences Building, 24 Tyndall Avenue, Bristol BS8 1TQ, UK. ✉email: i.sparkes@bristol.ac.uk

Mitochondria play essential roles in several metabolic pathways including photorespiration, biosynthesis of coenzymes and vitamins, and are required for cellular respiration[1–3]. Mitochondria number is carefully controlled through changes in fission and fusion rates[4,5]. The molecular components required for fission have been identified, whereas fusion mutants have proved more elusive. Pleotropic defects of Arabidopsis mitochondrial fission mutants (including slower growth and altered biomass) highlight a crucial role for the regulation of mitochondrial morphology in plant development and growth[6–8]. Mitochondrial morphology is correlated with variations in energy supply, with higher availability favouring fission (fragmented) and higher demand favouring fusion (elongated)[9]. Therefore, fusion mutants that cannot generate elongated mitochondria may be unable to generate enough ATP to support growth resulting in lethality, which could explain the lack of identified fusion mutants.

Mitochondrial fission in yeast and animals is driven by dynamin proteins (Dnm1 and Drp1, respectively) and Fis1, which may act to recruit dynamins to the site of fission[4]. In higher plants, DRP3a/b (Drp1/Dnm1 orthologues) and Fis1a/b (Fis1 orthologues) are required, along with an independent PMD1/2 regulated pathway[6,8,10–14]. In parallel, the fusion machinery of animals and yeast is composed of GTPases in the inner (Opa1/Mgm1) and outer (Mfn1,2/Fzo1) mitochondrial membranes. However, to date there are no identified functional orthologues of the fusion machinery in higher plants[4]. The Arabidopsis Friendly mutant results in mitochondria which, at the ultrastructural level, appear to be tethered to one another[15]. It is unclear whether this represents a block in the separation of organelles post fission or docking prior to fusion. In *Physcomitrella patens*, MELL1 affects mitochondria biogenesis, but it is unclear if it positively regulates fusion or inhibits fission[16]. Therefore, whilst there appears to be some conservation in the fission machinery across eukaryotes, the fusion mechanism may be distinct and have evolved independently in higher plants[4].

Furthermore, in yeast and animals, the sites of fission and possibly fusion may be coordinated at the ER-mitochondrial interface[17,18]. Although a physical interaction between mitochondria and the ER has been suggested[9], the molecular components driving possible interactions in higher plants are unknown. In *Physomitrella patens*, overexpression of MELL1 was suggested to affect mitochondria interaction with the ER based on close association between the two organelles[16].

In *Sacchromyces cerevisae*, GTPase Gem1 affects mitochondria biogenesis through interaction with the ER[19]. In metazoans, the Gem1 homologues, Miro1 and Miro2, predominantly affect mitochondria motility[20]. Plants encode 3 isoforms of Miro with Miro1 mutants displaying elongated mitochondria in pollen tubes[21]. This mutant is embryonic lethal and is partially functionally redundant with Miro2[22]. These phenotypes are indicative of a disruption in mitochondrial biogenesis and provided the rationale to test whether Miro2 affects mitochondrial biogenesis possibly through interaction with the ER.

Here, we show that mitochondria are tethered to the ER in leaf epidermal cells and is regulated by AtMiro2. Constitutively active Miro2 (GTP locked) increases mitochondria size with decreased number indicative of changes towards fusion rates. Constitutively inactive AtMiro2 (GDP locked) decreases mitochondria size with increased number indicative of a change towards increased fission rates. The inactive form also decreases interaction of the mitochondria with the ER, suggesting that interaction may not be essential for mitochondrial fission.

## Results

### Mitochondria are tethered to the ER in leaf epidermal cells.
In a similar manner to many organelles, mitochondria appear closely associated to the ER (Supplementary Movie 1). Correlations between movement of organelles in a densely populated cytoplasm may relate to random 'collisions' rather than regulated events between tethered organelles. To test whether mitochondria are tethered to the ER, we used optical tweezers to trap mitochondria and assessed whether subsequent movement dragged the ER in tobacco leaf epidermal cells. As the ER is a rapidly remodelling organelle, we performed these experiments in the presence of an actin depolymerising agent to inhibit overall organelle movement. We have previously used this principle to assess interactions between the ER–Golgi and peroxisome-chloroplast pairs[23,24]. Movement of trapped mitochondria reproducibly resulted in movement of trailing ER. Quantification indicated that 78 ± 5% remained attached with 22 ± 5% of trapped mitochondria detaching from the ER ($n = 64$) (Fig. 1 and Supplementary Movies 2, 3).

Considering ScGem1 affects mitochondria tethering to the ER, and HsMiro affects mitochondria motility, we sought to determine whether AtMiro2 affects either of these processes in leaf epidermal cells.

### AtMiro2 affects mitochondrial dynamics.
Miro2 is a large GTPase with two GTPase domains, two calcium-binding EF repeat domains and a C-terminal transmembrane domain (Supplementary fig. 1). Amino terminal fusions were generated to prevent masking the targeting information within the TMD[21]. GFP-full length AtMiro2 (GFP-AtMiro2) colocates to, and encircles, the inner mitochondrial membrane marker indicative of AtMiro2 being in the outer mitochondrial membrane marker (OMM) (Fig. 2b)). Morphological assessment indicated that GFP-AtMiro2 significantly increases mean mitochondrial area ($0.78 \pm 0.09 \ \mu m^2$ vs $0.54 \pm 0.03 \ \mu m^2$) and circularity ($0.83 \pm 0.02$ vs $0.77 \pm 0.02$), whereas the mean number of discrete mitochondria per cell is significantly decreased ($3.4 \pm 0.39$ vs $9.96 \pm 1.61$) when compared with cells expressing mitochondrial marker only (Fig. 2c, Supplementary fig. 2a). In addition, GFP-AtMiro2 significantly reduces mean mitochondrial speed compared with cells expressing marker alone ($0.49 \pm 0.03 \ \mu m \ s^{-1}$ vs $0.94 \pm 0.03 \ \mu m \ s^{-1}$, Supplementary movie 4 and 5, Fig. 2d, Supplementary fig. 2b). In comparison, Golgi speed was unaffected by GFP-AtMiro2 expression, indicating that the effect is specific to mitochondria ($0.98 \pm 0.015 \ \mu m \ s^{-1}$ control ($n = 1043$) compared with $0.99 \pm 0.02 \ \mu m \ s^{-1}$ GFP-AtMiro2 WT ($n = 730$) expressing cells, (data taken at 2 days expression from 29–31 cells, $p > 0.5$ students' $t$ test). Taken together these results suggest that Miro2 promotes mitochondrial fusion and inhibits motility. Furthermore, GFP-Miro2 can also result in clusters of mitochondria in a similar manner to overexpression of the mammalian fusion machinery component mitofusin 2[25].

We hypothesised that Miro2's effect on mitochondrial fusion was controlled by its GTPase domains. We generated mutations in conserved residues to mimic GTP and GDP-bound forms in both GTPase domains (Fig. 3) and compared morphological parameters and motility between mutants and with GFP-Miro2 WT. Rationale for the mutations was based upon previous studies that have shown point mutations that are sufficient for rendering the GTPase domains of Miro orthologues either constitutively active (GTP-'locked') or inactive (GDP-'locked')[26–28]. AtMiro2 KKVV (K23V and K434V) relates to GTP bound, whereas AtMiro2 SSNN (S28N and S439N) refers to GDP-bound form (Fig. 3).

Expression of the mutants did not affect the subcellular localisation of AtMiro2 (Fig. 3b, c), but there were observable differences in mitochondrial morphology and motility. Mitochondrial circularity remained unaffected by either AtMiro2

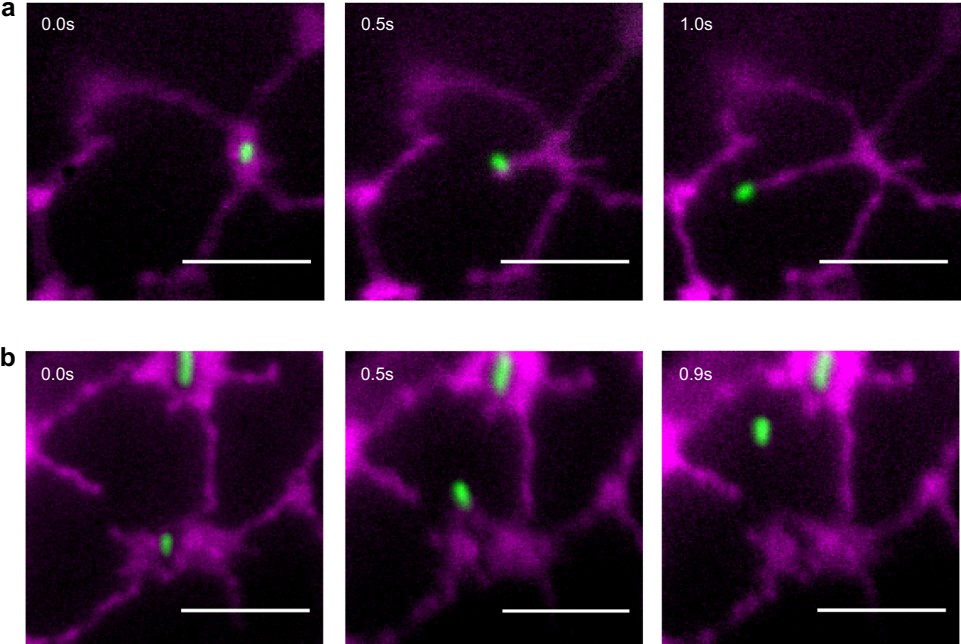

**Fig. 1 Mitochondria are physically attached to the ER.** Consecutive TIRF microscopy images showing ER (magenta) either following **a** or separating **b** from a mitochondrion (mito, green) that has been optically trapped and moved using an automated routine; trapping laser power of 40 mW, moved 6 μm at 6 μm s$^{-1}$. Scale bar indicates 5 μm, time is indicated in seconds (s). Cells were treated with latrunculin b.

mutant when compared with AtMiro2 WT or to one another. In cells expressing GFP-AtMiro2 SSNN, there was a significant increase in the average number of mitochondria per cell area (8.31 ± 1 vs 3.42 ± 0.4) with a significant decrease in mean area (0.42 ± 0.02 μm$^2$ vs 0.78 ± 0.09 μm$^2$) compared with cells expressing GFP-AtMiro2 WT (Fig. 3d, Supplementary fig. 3a). These results are suggestive of increased fission when Miro2 is inactive. Comparisons between GFP-AtMiro2 KKVV and GFP-AtMiro2 SSNN indicate significant changes in mitochondrial area and number per cell area, with AtMiro2 KKVV favouring larger fewer mitochondria (Fig. 3d). Results are suggestive of active Miro2-promoting mitochondrial fusion. Expression of GFP-AtMiro2 KKVV resulted in no significant changes in mitochondria area (0.6 ± 0.06 μm$^2$) or number (4 ± 0.46) when compared with GFP-AtMiro2 WT (Fig. 3d), suggestive of the wild type form being predominantly bound to GTP. Interestingly, clusters of mitochondria were observed in cells expressing either GFP-AtMiro2 WT or GFP-AtMiro2 KKVV. These results are suggestive that the GTPase domain of Miro2 when active favours mitochondrial fusion, with fission dominating when inactive.

In cells expressing either GFP-AtMiro2 SSNN (0.91 ± 0.03 μm s$^{-1}$, Supplementary movie 6) or GFP-AtMiro2 KKVV (0.7 ± 0.05 μm s$^{-1}$, Supplementary movie 7), mitochondria speed is significantly higher than in cells expressing GFP-AtMiro2 WT (0.49 ± 0.03 μm s$^{-1}$, Supplementary movie 5, Supplementary fig. 3b). In addition, movement in GFP-AtMiro2 SSNN-expressing cells is significantly faster than in those expressing GFP-AtMiro2 KKVV. Mitochondrial speed in GFP-AtMiro2 SSNN-expressing cells is similar to control cells expressing mitochondrial marker only (0.94 ± 0.03 μm s$^{-1}$, Supplementary movie 4). These results are indicative of Miro2 effecting mitochondrial dynamics in a similar manner to that in metazoans.

**AtMiro2 GTPase status affects ER-mitochondrial tethering.** To interrogate the mechanism governing Miro2's action on mitochondrial biogenesis, using optical tweezers, we tested whether mitochondrial tethering to the ER was affected. There was a significant increase in the number of mitochondria attached to the ER in the presence of the AtMiro2 WT (89.06 ± 3.66%) and the KKVV variant of AtMiro2 (85.94 ± 3%) compared with SSNN (72 ± 3%, Fig. 4, Supplementary fig. 4). AtMiro2 WT and KKVV mitochondria are therefore more tightly or more frequently connected to the ER in leaf epidermal cells, whereas AtMiro2 SSNN mitochondria are less tightly or less frequently connected to the ER. This suggests that the GTPase activity of AtMiro2 has a regulatory role in ER–mitochondria interaction in plant cells, with attachment favouring mitochondrial fusion.

## Discussion

In summary, our results support the model that Miro2 actively promotes mitochondrial fusion through interaction with the ER in tobacco epidermal cells. When inactive the cell attempts to maintain the chondriome volume through increasing fission, which seems to be independent of physical tethering to the ER. More recently, it has been suggested that mitochondrial fission rates may be controlled through mechanical encounters with the ER which effectively 'squeeze' mitochondria through a confined space as it passes over the ER[29]. Here, the role of the ER in this process is indirect. We hypothesise therefore that when mitochondrial movement is increased in the inactive/GDP-bound Miro2 mutant, organelles pass over confined zones occupied by the ER more frequently triggering increased fission of mitochondria. Increase in speed in the inactive variant may be due to effectively 'releasing' mitochondria from interaction with the ER so that the organelles are able to move more 'freely' unhindered through attachment to the large ER network. In metazoans, Miro interacts with TRAK proteins, kinesin and dyneins to drive microtubule-dependent motion. However, it now appears that HsMiro may regulate mitochondria movement through both microtubule and actin-dependent processes[30]. Considering that movement in higher plants is predominantly actin-dependent leads to the interesting suggestion that Miro2 may, akin to HsMiro1/2, recruit myosin molecules to regulate movement[31].

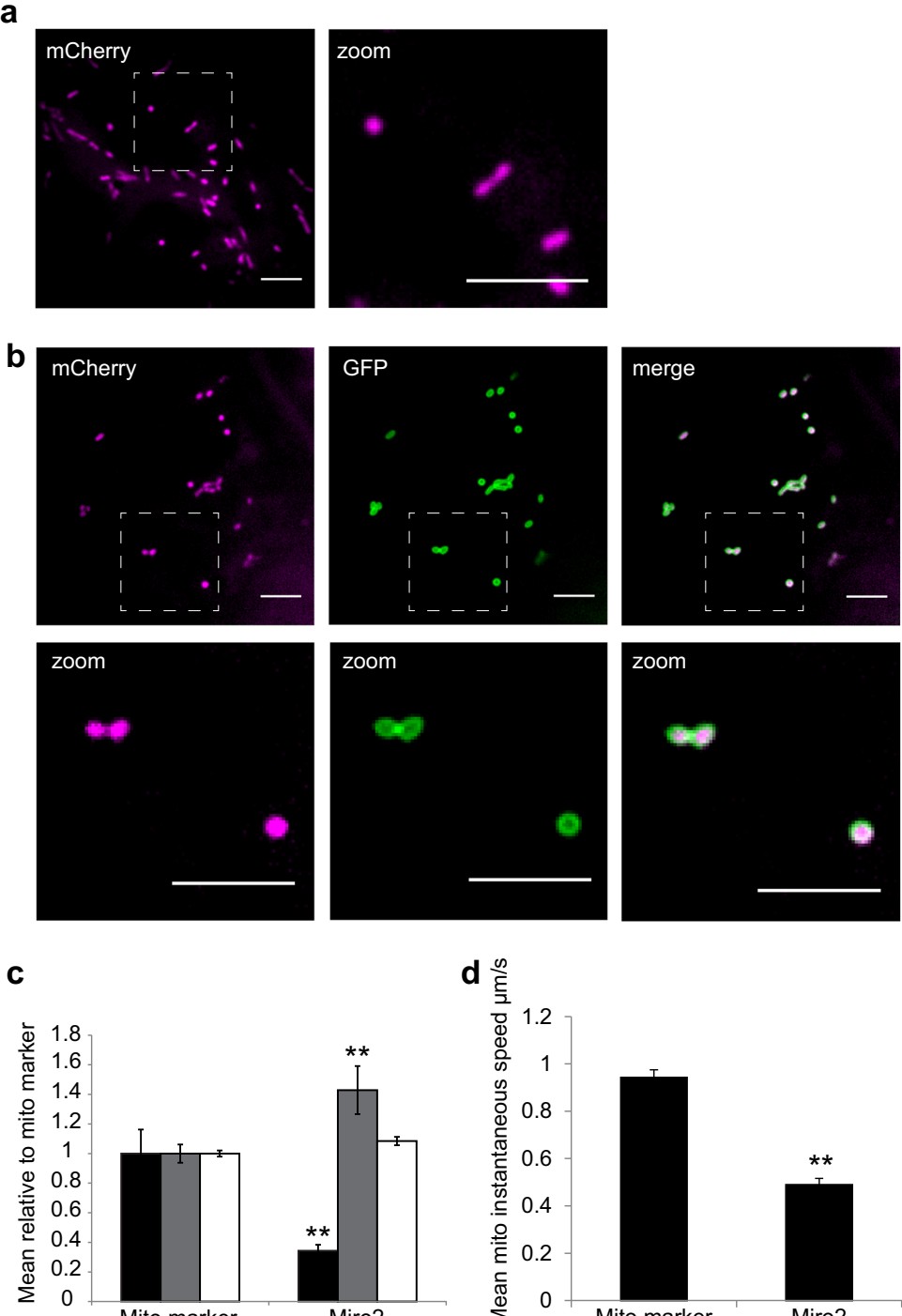

**Fig. 2 At Miro2 colocates to mitochondria and affects their morphology and dynamics. a–b** Spinning disc confocal microscopy images of tobacco leaf epidermal cells transiently expressing the mitochondrial inner membrane marker alone (**a**, magenta), or with GFP-AtMiro2 (**b**, green). Hatched box overlay indicates the area displayed in zoom below. Scale bar is 5 μm. **c** Bar chart showing mean and SEM for mitochondria number per cell region (black bar), area (μm$^2$, grey bar) and circularity (white bar) in cells coexpresing GFP-AtMiro2 WT with mitochondria marker, relative to cells expressing mitochondria marker only. For the mitochondria marker only sample, $n = 229$ from 24 cells. GFP-AtMiro2-expressing cells, $n = 89$ from 26 cells. **d** Bar chart showing mean with SEM for average mitochondrial speeds (μm s$^{-1}$). The movement of individual mitochondria was tracked in 24–26 cells with a total of 770 marker only organelle tracks and 120 AtMiro2 organelle tracks. Data for **c** and **d** were taken from three independent experiments and analysed using a $t$ test with Welche's correction, **$p < 0.02$. Individual data-points for **c** and **d** are plotted in Supplementary fig. 2.

Future studies are required to define myosin recruitment to mitochondria and whether subsequent movement is controlled through co-regulation of myosin action exerted on the mitochondria and uncoupling of the 'opposing' force tethering the organelles to the ER.

Mitochondrial fusion is controlled by outer mitochondrial membrane (OMM) proteins mitofusin 2 (MFN2) and Fzo1 in mammals and yeast, respectively[32–34]. Similar to our results presented for AtMiro2, MFN2's GTPase domain is critical for mitochondrial fusion[35], and it regulates ER–mitochondria

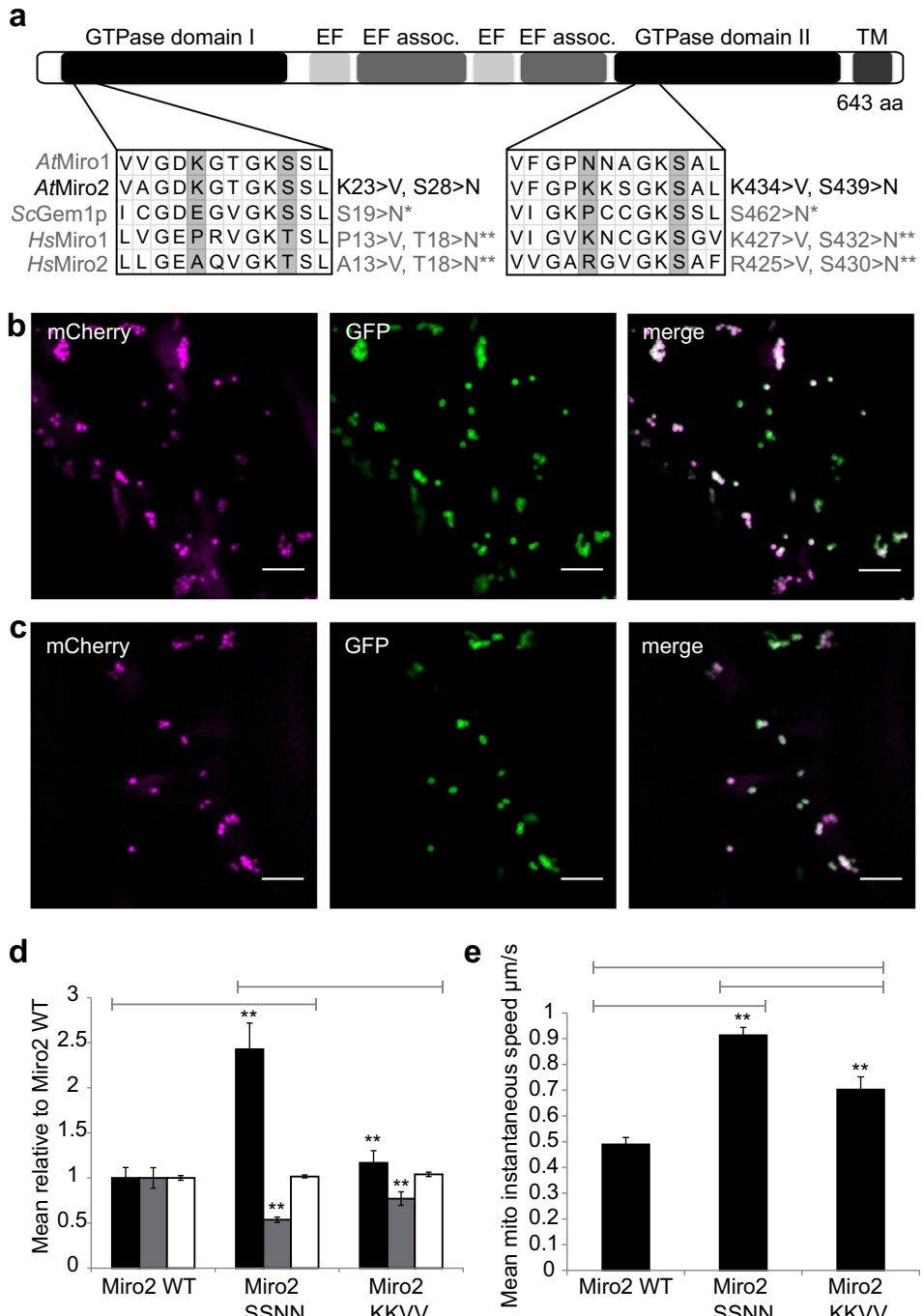

**Fig. 3 Mutation of AtMiro2's GTPase domain does not affect mitochondrial targeting but does affect mitochondrial morphology and motility.**
**a** Schematic showing domain organisation of *A. thaliana* Miro2. The location of the single amino acid mutations introduced into each GTPase domain are indicated, as is an alignment with the corresponding residues in *A. thaliana* Miro1, *S. cerevisiae* Gem1p and *H. sapiens* Miro1 and Miro2 and the corresponding orthologous functional GTPase mutations as previously published[27,28,30]. **b–c** Spinning disc confocal images of tobacco leaf epidermal cells transiently coexpressing mitochondria inner membrane marker (magenta) with GFP-AtMiro2 SSNN (green, **b**), or GFP-Miro2 KKVV (green, **c**). Scale bar, 5 µm. **d** Bar chart showing mean and SEM for mitochondria number per cell region (black bar), area (µm$^2$, grey bar) and circularity (white bar) in the mutants relative to AtMiro2 WT mitochondria. **e** Bar chart showing average mitochondria speeds (µm s$^{-1}$). Analysis for **d** and **e** are derived from 24 and 26 cells from three independent experiments. For the morphological analysis in **d**, GFP-AtMiro2 $n = 89$, GFP-AtMiro2 SSNN $n = 216$, GFP-AtMiro2 KKVV $n = 104$ organelles. For movement analysis in **e**, GFP-AtMiro2 $n = 120$, GFP-AtMiro2 SSNN $n = 566$, GFP-AtMiro2 KKVV $n = 149$ organelle tracks. Data were analysed using a *t* test with Welche's correction, **$p < 0.01$. Individual data-points for **d** and **e** are plotted in Supplementary fig. 3.

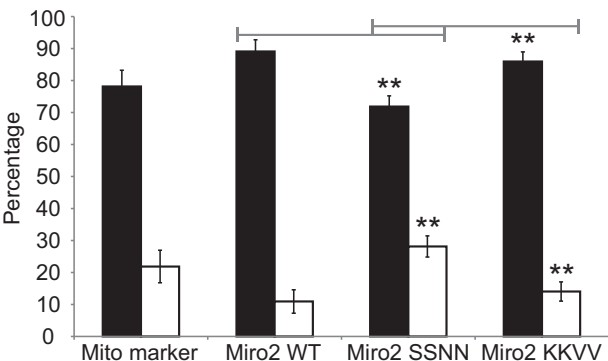

**Fig. 4 Mitochondrial tethering to the ER is affected by AtMiro2.** Bar chart showing the mean percentage of events: endoplasmic reticulum (ER) was observed to either follow (black), or detach from (white), an optically trapped mitochondrion. In cells expressing either mCherry mitochondria marker, AtMiro2 WT, SSNN, or KKVV with an ER marker, an automated routine (maintaining a set trapping laser power of 40 mW, a set distance of 6 μm and a set speed of 6 μm s$^{-1}$) was used to trap and move 64 independent mitochondria from 9 to 12 biological replicates from three independent experiments. Cells were treated with latrunculin b. Error bars are SEM. Data were analysed using a Welche's *t*-test of weighted means, **p < 0.01. Individual data-points are plotted in Supplementary fig. 4.

tethering, however the exact role in the latter is controversial[36,17]. Our results are therefore similar to MFN2 and we propose that in higher plants, AtMiro2 is a functional orthologue of MFN2.

In addition to affecting mitochondrial fusion, we show that AtMiro2 affects mitochondria motility. Members of the Miro gene family display different functional characteristics; ER–mitochondria tethering in yeast (ScGem1) and affecting mitochondria motility in mammals (HsMiro1). Therefore, in higher plants it appears as though AtMiro2 has evolved both of these functional characteristics[20].

Isolation of molecular components involved in mitochondrial fusion have proved elusive. Potentially, considering energy demand is likely correlated with fusion events, mutants defective in fusion may be unable to generate enough ATP to sustain growth resulting in lethality. Interestingly *Atmiro1* knockouts are embryo lethal with pollen tubes displaying elongated mitochondria. Given the partial functional redundancy between Miro1 and Miro2, both GTPases may act to control mitochondria biogenesis. Considering the potential issue with isolating viable fusion mutants using a forward genetics approach, future proteomic studies of AtMiro2 variants (GTP/GDP locked) may provide useful insight into the molecular machinery involved in regulating mitochondria fusion. Potential interactors could include homologues of interactors involved in the MFN2 dependent pathway.

Future protein interaction studies will enable us to determine how Miro2 fits in with the currently identified fission machinery, and how altering tethering with the ER impacts on molecule exchange between the two organelles. This is likely to have a large impact on plant physiology given conserved roles in lipid and calcium exchange highlighted in yeast and mammals[37,38]. In a broader context, mitochondria and ER movement and interaction may provide important adaptive responses to environmental stresses including exchange during plant defence[39–42].

## Methods

**Cloning and mutagenesis of AtMiro2 GFP fusions**. AtMiro2 cDNA (LIC6 vector containing the coding sequence for AT3G63150) was purchased from the Arabidopsis Biological Resource Centre. The full AtMiro2 coding sequence was cloned using Gateway reactions (Invitrogen) into the binary destination vector PB7WGF2 creating an N-terminal GFP fusion. Using this as template, the Quickchange site-directed mutagenesis kit (Agilent Technologies) was used according to the manufactures' instructions to sequentially introduce point mutations into both GTP-binding domains, to create KKVV (K23>V and K434>V) and SSNN (S28>N and S439>N) double mutants.

**Plant material and sample generation**. *Nicotiana tabacum* (tobacco) plants were grown and transiently transformed using Agrobacterium infiltration method[43]. Competent, transformed Agrobacteria containing GFP-AtMiro2 (WT, SSNN and KKVV) constructs, or fluorescent fusions targeted to inner mitochondrial mitochondria membrane (MT-mCherry, MT-GFP) or the ER lumen (ER-mCherry)[44] were infiltrated into leaf tissue at an optical density of 0.05. Leaf samples (~5 mm$^2$) were taken for immediate analysis from plants following 2 days of expression.

**Spinning disc confocal imaging**. Spinning disc confocal imaging of mitochondria (MT_rk or MT_gk), GFP-Miro2 fusions and ER (ER_rk) in live tobacco epidermal pavement cells was performed using a VisiScope Confocal Cell Explorer under the control of VisiView software (Visitron Systems, GmbH Germany), composed of an IX81 motorised inverted microscope (Olympus, Germany), a CSU-X1 Spinning Disc unit (Yokogawa, Japan), a PlanApo UPlanSApo ×100 (1.4 NA) oil objective (Olympus, Germany) with a Photometrics CoolSNAP HQ2 camera (Roper Scientific, Germany). To achieve dual fluorescent imaging, GFP was excited with a Sapphire 488 nm 70 mW laser and mCherry with a Cobolt Jive 561 nm 70 mW laser. All movies were taken using a temporal resolution of five frames s$^{-1}$, 100 frames long with a spatial resolution of 0.129 μm pixel$^{-1}$.

**Mitochondria number, morphology and movement quantification**. Spinning disc movies of individual leaf epidermal cells were taken as described above. Within these movies a fixed region (17.8 μm × 13.9 μm) was used to normalise the area of each cell analysed. The mCherry signal of mitochondria was automatically detected using a graphical user interface developed in-house, coded in Matlab R2014b, which identifies and tracks mitochondria as objects after Gaussian filtering and thresholding, using the tracking algorithm developed by Crocker et al.[45]. Average speeds of individual mitochondria were calculated. Single-frame images from these time lapse movies were used to analyse mitochondria within the same defined region using an ImageJ macro developed in-house. Mitochondria were positively identified using the mCherry marker signal and, after Gaussian filtering, thresholding and segmentation, a number of parameters within were calculated: (1) the number of mitochondria per unit cell area, (2) the surface area of each mitochondria and (3) the perimeter of each mitochondria. Values from 2 and 3 were used in the following equation to calculate the circularity of each mitochondria: circularity $= \frac{4\pi \text{ surface area}}{\text{perimeter}^2}$. The mean and standard error of the mean was calculated using data collected over three biological replicates for all of the quantification carried out. Statistical tests were performed as indicated.

**Optical trapping setup, data generation and analysis**. An Elliot scientific optical trapping platform was fitted to a Nikon Ti-U inverted microscope adapted for two-channel TIRF microscopy (TIRF-M). A 1090 nm trapping laser was delivered to the trapping objective; ×100 NA 1.49, oil immersion with both temperature and cover glass correction ring with a further ×2.5 magnifier optics before the EMCCD cameras. An Omicron laser hub fibre coupled to the manual Nikon TIRF-M unit delivered excitation wavelengths for GFP (488 nm) and RFP (561 nm). Two Electron Multiplying Charge-Coupled Devices (EMCCD, iXon, Andor) cameras connected via a twin-cam (Cairn) unit detected fluorescence emissions. Microscope stage control, the trapping laser shutter, TIRF angle control and image acquisition were controlled by a custom software developed using National Instruments LabVIEW 2012 with the Vision package.

Bead trapping profiles were generated each day and the power at the sample calibrated to 40 mW. Raw images of the automated trapping routine were recorded from time 0 of the stage movement, acquired from the synchronised EMCCDs as 16-bit TIFF stacks. The automated trapping routine was programmed as follows: the trap was turned on after 1 s, the stage was moved (translation) 6 μm at 6 μm s$^{-1}$ between 5 s and 6 s, the trap was turned off after 7 s and image capture ended at 18 s. (0.1 s frame rate, i.e., 10 frames per seconds). Organelles which remained trapped and moved the entire 6 μm distance were observed to determine whether they remained attached or detached from the ER. Statistical analysis used weighted means per sample as the number of trapped organelles per sample varied.

**Chemicals and probes**. For actin depolymerisation, leaf samples were incubated for 60 min prior to imaging in 25 μM latrunculin b (Merck Millipore) in water made from a 10 mM DMSO stock.

**Statistics and reproducibility**. Data were generated from independent experiments, defined as independent infiltrations, with samples taken from several plants. Samples sizes are as indicated in the figure legends and as follows: mitochondrial morphological analysis of control (24 cells $n = 229$), AtMiroWT (26 cells, $n = 89$), AtMiro2 SSNN (26 cells, $n = 216$), AtMiro2 KKVV (26 cells, $n = 104$) was analysed using a Welche's *t*-test; movement analysis of control (24 cells, $n = 770$), AtMiroWT (26 cells, $n = 120$), AtMiro2 SSNN (26 cells, $n = 566$), AtMiro2 KKVV

(26 cells, $n = 149$) was analysed using a Welche's *t*-test; optical trapping data ($n = 64$) was analysed using Welche's *t*-test of weighted means.

**Reporting summary**. Further information on research design is available in the Nature Research Reporting Summary linked to this article.

## Data availability

All data generated or analysed during this study are included in this published article (and its supplementary information files). The source data behind the graphs are available in Supplementary Data 1.

## Code availability

Algorithms generated for this study are available at https://github.com/congping/mito_tracker.

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

## Acknowledgements

We thank professor Michael Schrader for insightful discussion. This work is funded by grants from the Leverhulme Trust (RPG-2015-106) and STFC (PM-1216) awarded to I.S.

## Author contributions

I.S. conceived the experiments and oversaw the project. I.S. and R.R.W. wrote the manuscript with input from co-authors. R.R.W. generated all the raw data relating to

protein localisation and mitochondrial dynamics and generated and analysed the optical trapping, mitochondrial morphology. R.R.W., I.G.C., I.L. generated the clones. A.W., B.C.B., S.B. developed and facilitated use of the optical tweezer system. J.M. automated the size measurement algorithm for mitochondria. C.L. developed the tracking programme with P.A. and analysed mitochondria dynamics. I.L. generated the data and quantified effects on Golgi movement.

## Competing interests

The authors declare no competing interests.
