## [Peer Review File · Communications Biology]

Reviewers' comments:

Reviewer #1 (Remarks to the Author):

This paper describes that the Arabidopsis Miro2 protein is involved in the mitochondrial biogenesis, probably through fusion, and interaction between ER and mitochondria. The manuscript is well written and figures are also well organized, so I did not notice the typos or imperfections. Heterologous (Arabidopsis gene in tobacco), transient and over-expression with fused fluorescent protein might be problematic, but the imaging data and optical tweezer data are impressive and straightforward. If it is possible, addition of data with non-GFP fused AtMiro2 in the Arabidopsis T-DNA tagline (or CRISPRed KO lines) will be suitable for the clear and genuine understanding.

Reviewer #2 (Remarks to the Author):

The paper focuses on the GTPase Miro2, which is a homolog of GEM1, a protein involved in mitochondrion biogenesis at the ER. The authors overexpress GFP-tagged AtMiro2 in tobacco leaf cells to determine Miro2 localization. GFP-tagged Miro2 overexpression yields a mitochondrion number, morphology and movement speed phenotype, which provides clues to the function of Miro2. Further experiments using a dominant negative and constitutively active Miro2 have been performed, suggesting that active Miro2 promotes mitochondrial fusion, while inactive Miro2 results in mitochondrial fission. Since Miro2 activity is linked to the ER, the authors test the effects of the GFP-tagged overexpression mutants on the connection strength of the mitochondria with the ER using elegant optical trapping experiments. The authors conclude that the GTPase activity of Miro2 is important for mitochondria-ER connectivity.

The paper is well written and carefully worded. The research is interesting and potentially relevant for a broad range of cell biological research. The experiments would be more convincing with a Miro2 mutant, but given that Miro2 is partially functionally redundant with the embryo-lethal Miro1, that might be unfeasible and out of the scope of this story.

Major points:

- However carefully worded, all the experiments are carried out using overexpression of a sub-functional GFP-tagged protein from a heterologous system. This makes interpretation of the results difficult. It is unclear whether generation of a mutant (by CRISPR or otherwise) has been attempted. The results would be more convincing in a mutant background using overexpression of a protein from a non-heterologous system, but are acceptable if the generation of a mutant proves unfeasible.

Minor points:

- Line 142: Clusters of mitochondria are not observed in either GFP-AtMiro2 WT or GFP-AtMiro2 KKVV, but in both.
- Line 168: The authors claim that the GTPase activity of Miro2 plays an important role in ER-mitochondria interaction in plants. However, the difference in ER-attachment between the WT and GDP-locked Miro2 mutant is only 17%, with still 72% of mitochondria attached to the ER in the latter. Therefore, this claim is overstated.

Reviewer #3 (Remarks to the Author):

This is a well-written and interesting short MS. The overall quality of the data is high while the images

and moves are impressive. Interestingly, image analysis with wt and mutated fusions in transgenic cells show a unique function of Miro2 in promoting Mt fusion via ER interaction. I would support its publication even though it lacks biochemical data--which in the revision the authors could provide a more detailed discussion in this aspect for the underlying mechanism in future study.

Reviewer 1

We are pleased with the reviewer's positive response towards the manuscript. Please find below our response to their query shown in italics. In addition, please note that the manuscript has been altered to comply with the Comms Biol format.

If it is possible, addition of data with non-GFP fused AtMiro2 in the arabidopsis T-DNA tagline (or crisprized KO lines) will be suitable for the clear and genuine understanding

AtMiro2 T-DNA insertional mutants were reported to be viable, although may express a truncated variant consisting of the functional GTPase domains (Yamaoka and Leaver, Plant Cell 2008). Given the partial functional redundancy between Miro1 and Miro2, it is unclear if a *miro2* ko, if viable, would provide 'clarity' on the role of Miro2 as it may be partially restored through the action of Miro1. Considering *miro1* knockout is embryo lethal, a *miro1 miro2* knockout would not be viable.

We have attempted to generate transgenic Arabidopsis lines expressing the wild type, GDP or GTP locked variants of Miro2 (used in the manuscript) in a Columbia background. We are unable to generate viable homozygous GDP locked lines. Our results indicate that expression of the constitutively inactive (GDP) variant is lethal. Therefore, even if it were possible to isolate / generate *miro2* ko, it would be impossible to assess the function of the constitutively inactive (GDP) form as the progeny would not be viable.

Reviewer 2

We are pleased with the reviewer's positive response towards the manuscript. Please find below our response to their queries shown in italics. In addition, please note that the manuscript has been altered to comply with the Comms Biol format.

The experiments would be more convincing with a Miro2 mutant, but given that Miro2 is partially functionally redundant with the embryo-lethal Miro1, that might be unfeasible and out of the scope of this story.

The results would be more convincing in a mutant background using overexpression of a protein from a non-heterologous system, but are acceptable if the generation of a mutant proves unfeasible.

AtMiro2 T-DNA insertional mutants were reported to be viable, although may express a truncated variant consisting of the functional GTPase domains (Yamaoka and Leaver, Plant Cell 2008). Given the partial functional redundancy between Miro1 and Miro2, it is unclear if a *miro2* ko, if viable, would provide 'clarity' on the role of Miro2 as it may be partially restored through the action of Miro1. Considering *miro1* knockout is embryo lethal, a *miro1 miro2* knockout would not be viable.

We have attempted to generate transgenic Arabidopsis lines expressing the wild type, GDP or GTP locked variants of Miro2 (used in the manuscript) in a Columbia background. We are unable to generate viable homozygous GDP locked lines. Our results indicate that expression of the constitutively inactive (GDP) variant is lethal. Therefore, even if it were possible to isolate / generate *miro2* ko, it would be impossible to assess the function of the constitutively inactive (GDP) form as the progeny would not be viable.

- Line 142: Clusters of mitochondria are not observed in either GFP-AtMiro2 WT or GFP-AtMiro2 KKVV, but in both.

We respectfully suggest the reviewer reread the line as there appears to be some confusion (Line 142 in the original manuscript, now line 158)

- Line 168: The authors claim that the GTPase activity of Miro2 plays an important role in ER-mitochondria interaction in plants. However, the difference in ER-attachment between the WT and GDP-locked Miro2 mutant is only 17%, with still 72% of mitochondria attached to the ER in the latter. Therefore, this claim is overstated.

We have replaced the word 'important' with 'regulatory' (Line 168 in the original manuscript, now line 179).

Reviewer 3

We are pleased with the reviewer's positive response towards the manuscript. Please find below our response to their query shown in italics. In addition, please note that the manuscript has been altered to comply with the Comms Biol format.

I would support its publication even though it lacks biochemical data--which in the revision the authors could provide a more detailed discussion in this aspect for the underlying mechanism in future study.

We have extensively revised the discussion section (text highlighted in yellow) to include a more comprehensive discussion relating to the mechanism controlling mitochondrial fusion and movement. Comparisons between AtMiro2 and components involved in mitochondrial fusion in yeast and mammals are included, and we propose that AtMiro2 is a functional orthologue of HsMFN2.

REVIEWERS' COMMENTS:

Reviewer #1 (Remarks to the Author):

This paper describes the mitochondrial fusion and interaction with ER, and the analysis of Miro2 in these phenomena. Revision is mainly on introduction and discussion parts and I could not find any inappropriate parts in them. I think that the results have been good enough to be published.

Reviewer #2 (Remarks to the Author):

After reviewing the manuscript again, I conclude that the authors have answered all my concerns, and no new concerns have arisen.